# Association of Food Consumption According to the Degree of Processing and Sociodemographic Conditions in Older Adults

**DOI:** 10.3390/foods12224108

**Published:** 2023-11-13

**Authors:** Graziele Maria da Silva, Daniela de Assumpção, Carolina Neves Freiria, Flávia Silva Arbex Borim, Tábatta Renata Pereira de Brito, Ligiana Pires Corona

**Affiliations:** 1Faculdade de Ciências Aplicadas, Universidade Estadual de Campinas, Street Pedro Zaccaria, 1300, Limeira 13484-350, São Paulo, Brazil; licorona@unicamp.br; 2Faculdade de Ciências MédicasUniversidade Estadual de Campinas, Street Tessália Vieira de Camargo, 126-Cidade Universitária, Campinas 13083-887, São Paulo, Brazil; danideassumpcao@gmail.com (D.d.A.); carolnfreiria@gmail.com (C.N.F.); flarbex@hotmail.com (F.S.A.B.); 3School of Nutrition, Federal University of Alfenas, Street Nabor Toledo Lopes, 598-Parque das Nações, Alfenas 37130-000, Minas Gerais, Brazil; tabatta_renata@hotmail.com

**Keywords:** aging, NOVA classification system, educational status

## Abstract

Several factors can impact food consumption in older adults, including those of sociodemographic, physiological, and chronic non-communicable diseases. This study aimed to evaluate the association of food consumption according to its degree of processing with sociodemographic conditions in community-dwelling older adults. Food intake was evaluated from 24-h recall data. All food items were classified according to the degree of processing into four groups as follows: in natura or minimally processed, culinary ingredients, processed, and ultra-processed foods. Food groups were considered dependent variables in a quantile regression model, adjusting for sex, age, schooling, ethnicity, and number of residents. Women and individuals with higher levels of education had lower consumption of in natura or minimally processed foods and higher consumption of ultra-processed foods. The yellow or indigenous ethnicity presented the lowest consumption of processed foods; older people who lived with three or more individuals had the highest consumption of culinary ingredients, whereas the older people who lived with one to two people had the highest consumption of processed foods and the lowest consumption of ultra-processed. These groups may be the target of educational and public policies to improve diet quality and contribute to quality of life in older ages.

## 1. Introduction

Several factors, including social and environmental factors, purchasing power, and lack of social support for the purchase and preparation of food, can affect food consumption in older adults [1]. The highest morbidity and mortality rates are often observed among individuals with worse socioeconomic conditions [2]. A study evaluating the social determinants of health in older adults, such as income, education, occupation, family structure, and exposure to diseases, among others, has demonstrated that the occurrence of chronic non-communicable diseases (NCDs) increased owing to demographic changes with populational aging, and poverty and social exclusion were the aggravating factors for this morbidity [1].

Education level is often an important predictor of healthier eating habits [3], as less educated individuals may make choices considered inadequate owing to lack of access to healthy foods, food prices, and difficulties related to selection and preparation, among other factors. A study that evaluated sociodemographic factors related to mortality in 12,373 older adults (≥65 years) in Latin America, India, and China has reported that lower education, lower professional achievement, assets, and receipt of pension were associated with higher mortality and food insecurity in this population. The study also emphasizes that the main causes of death were attributed to NCDs, especially stroke [4].

In Brazil, food choices in this population have changed because of changes in dietary patterns. Among the main dietary modifications observed are the consumption of foods of high energy density and rich in fats and sugars and the decrease of basic items, such as rice and beans, which is typical of the Brazilian diet [5]. Data from the National Health Survey (PNS, 2019) revealed that more educated older adults were 80% more likely to consume vegetables, fruits, and milk; however, they were 50% less likely to consume meats and beans compared with less educated individuals. The study concluded that despite these changes, less educated individuals would have the most inadequate food consumption [6].

Thus, to elucidate the reality and food determinants in public health, studies of food consumption have been conducted according to the NOVA classification, a method that can assess the intake of all food groups divided into in natura or minimally processed, culinary ingredients, processed, and ultra-processed food items [7]. The Food Guide of the Brazilian Population, published in 2014, adopted the recommendation of food consumption by the NOVA classification because it states that disease prevention occurs through combinations of nutrients and other chemical compounds that are part of the food matrix, more than isolated nutrients. In addition, it states that to promote healthy eating, understanding a set of strategies aimed at providing individuals and collectivities with the realization of appropriate eating practices is necessary [8].

A study, which evaluated the trend of food acquisition according to the NOVA classification in Brazil, using *Pesquisa de Orçamentos Familiares* (POF—Brazilian Family Budgets Survey), data from 1987–1988 to 2017–2018, has reported that although the food consumption of Brazilians still has a higher prevalence of in natura or minimally processed foods and culinary ingredients, a trend of increase of 0.4 percentage points annually has been observed for the participation of ultra-processed foods in the diet from 2002 to 2009 and 0.2 percentage points from 2008 to 2018. This trend was higher in those with higher incomes and those who lived in the south and southeast urban and metropolitan regions of the country [9].

The high consumption of ultra-processed foods has been associated with an increased risk of chronic NCDs, such as overweight and obesity. This is caused by the fact that these foods are of low nutritional quality and have high concentrations of fats, salt, sugar, and chemical additives. Data from the published report of the Pan American Health Organization in 2015 revealed that the sale of ultra-processed foods increased in the 13 countries analyzed, and this consumption was associated with an increase in overweight and obesity in the population [10]. Other studies have reported that the consumption of ultra-processed foods in middle-aged adults (mean, 55 years) and older adults (>60 years) was also associated with hypertension and cardiovascular diseases [11], diabetes [12], cancer [13], frailty syndrome [14], and dementia [15].

However, few studies evaluated the food consumption of older adults according to the degree of NOVA classification. Moreover, most of the studies published on this subject evaluated only the consumption of ultra-processed foods and their impacts on health without considering the intake of other food groups [16,17].

An adequate diet is fundamental for the promotion and maintenance of health in aging, and as social factors are very important in the interface between nutrition, development of diseases, and maintenance of health, studies evaluating food intake in aged populations are very important to allow health educational strategies for older people in general, and particularly for those with lower education levels who tend to have less information regarding health and nutrition. Therefore, this study aimed to evaluate the association of food consumption according to its degree of processing with sociodemographic conditions in community-dwelling older adults

## 2. Materials and Methods

This is a cross-sectional study with a convenience sample of 611 older adults from three Brazilian cities: Campinas, Limeira, and Piracicaba, with data collected from October 2018 to December 2019. In each municipality, basic health units (BHU) participating in the Family Health Strategy (FHS) were indicated by the respective health department of each municipality, representing all regions. In each BHU, the health team invited people who met the inclusion criteria to participate in the research. Details on the sample size and collection procedures can be found in previous publications [18].

The inclusion criteria were the age of ≥60 years, residing in one of the participating cities, registered in the FHS, and with a minimum capacity to understand the study procedures and the consent form without the need for an auxiliary informant. The exclusion criteria were the use of dietary supplements based on vitamins and/or minerals (since the main study aimed to assess the deficiency of some of these nutrients and their dietary consumption, not considering supplement use) and being monitored by a homecare program or receiving chemotherapy treatment since these conditions significantly alter food intake.

Of the 611 participants interviewed, 15 were excluded because they did not have complete data from the questionnaire and 20 because they did not have food consumption data. Therefore, our final sample for the present study comprised 576 older adults. The research protocol was approved by the Research Ethics Committee of UNICAMP in September 2018 under CAAE number 95607018.8.0000.5404. Data were collected at the BHU, where the participant registered and was followed up. The questionnaire contained items on personal data, sociodemographic conditions, health issues, and lifestyle. For this study, the following sociodemographic variables were used: sex (male and female); age (in years); schooling (0–4 years, 5–8 years, 9–11 years, and ≥12); marital status (with a partner and without a partner); ethnicity (white, black, brown, yellows, or indigenous) and number of residents (alone, 1–2, or ≥3).

### 2.1. Food Consumption

Food consumption was evaluated using a 24-h food recall (24hFR), applied by trained interviewers. The 24hFR is the most commonly used method in population research, being one of the main tools for quantification of food intake in addition to being a convenient method for application in individuals with illiteracy [19]. Several methodological precautions were adopted to minimize possible biases in food consumption assessment, as described in a previous publication [18]. Thus, we used photographic materials during the interview to help volunteers estimate the portions in order to minimize errors in reported quantity [20] and memory bias in the older adults [21]; no data were collected on Mondays to avoid capturing the eating patterns of the weekend that tend to be different from those of weekdays; and two rounds of data consistency were performed before quantification of reported portions into grams (g) and milliliters (ml). Subsequently, these amounts were entered into the Nutrition Data System for Research software (NDSR) version 2019 [22].

### 2.2. Statistical Analysis

For the construction of the food consumption variable according to its degree of processing, each food mentioned in the 24hFR was classified by two researchers of the study. Initially, the foods were grouped into subgroups comprising their equivalent foods. For example, banana, apple, grape, guava, etc., were grouped into “Fruits”. Subsequently, two codes were assigned to each food; the first corresponded to its equivalent subgroup, and the other was represented by its degree of processing, where the code “1” was composed of subgroups of in natura or minimally processed foods, “2” by subgroups of processed foods, “3” by subgroups of ultra-processed foods, and “4” by subgroups of culinary ingredients.

The NOVA classification was considered [7], dividing the foods into four groups. In natura or minimally processed included rice, legumes, meats (beef and pork), fruits, milk and natural yogurts (without added sugar), flours and pasta, chicken, roots and tubers, egg, fish, oilseeds, and others (coffee, tea, spices, and seeds). Culinary ingredients included oils, butter, and lard, with added sugar. Processed included French bread, cheeses, dried meats and bacon, others (pickled fruits and vegetables, tomato extract, oilseeds with salt, coconut milk, sweetened dried coconut, and garlic with salt). Ultra-processed included sweets and desserts, ultra-processed yogurts, ultra-processed breads, margarine, processed sausages and meats, salted snacks/biscuits, sweetened beverages, sweet cookies, soft drinks, commercial ultra-processed fat (hydrogenated vegetable fat, cream cheese, and curd), mayonnaise and sauces, and others (sweeteners, vanilla essence, baking powder, ready-made seasonings, soy extract, and sodium glutamate).

Next, the relative caloric contribution of each food in relation to the total calories consumed and the contribution of each of the four groups in relation to the total calories was calculated. The following subgroups of foods called “others” were excluded because they did not represent a considerable energy contribution value: alcoholic beverages, coffee, tea, spices, seeds, pickled fruits and vegetables, oilseeds with salt, coconut milk, sweetened dried coconut, garlic with salt, sweeteners, and vanilla essence. The percentages of the caloric contribution of the foods according to their degree of processing presented a non-normal distribution, according to the Shapiro–Wilk test of adherence to normality. Thus, non-parametric tests were applied to evaluate the differences between the variables.

To evaluate the differences in the means of the caloric percentages in relation to the categories of each of the sociodemographic variables, they were tested using the Mann–Whitney test for two independent samples, when the variable was binary, or by the Kruskal–Wallis test, when the variable presented three or more categories, then using Dunn’s post-hoc test. To evaluate the association between food consumption according to the degree of processing and sociodemographic variables, quantile regression models were developed, which aim to analyze the differences between the medians between the dependent and independent variables since the distribution was not normal. Each food group, according to its degree of processing (minimally processed foods, processed foods, ultra-processed foods, and culinary ingredients), was considered the dependent variable in each model, and for the independent variables, the variables of sociodemographic conditions were considered. All study analyses were performed using the STATA software, version 14, with a critical level of 5%.

## 3. Results

Most participants were women, aged between 60 and 64 years, had a partner, had 0–4 years of schooling, self-declared to be of white ethnicity, and lived with 1–2 people. The average total intake of the participants was 1607 kcal/day. In natura or minimally processed foods represented 61.2% of the energy consumed, followed by ultra-processed foods (16.3%), culinary ingredients (12.9%), and processed items (9.1%).

Table 1 presents the relative caloric contribution of each food group according to their degree of processing and sociodemographic conditions. The lowest consumption of the group of in natura or minimally processed foods was observed among women (*p* = 0.017), those with higher schooling (*p* = 0.002), and those without partners (*p* = 0.031).

Regarding culinary ingredients, older people who studied for ≥12 years (*p* = 0.020) were associated with lower consumption. For the group of processed foods, individuals who self-declared to be black or brown (*p* = 0.024) and yellow or indigenous (*p* = 0.036) had a lower consumption of processed foods. Older adults who studied from 9 to 11 years (*p* = 0.002) and ≥12 years (*p* = 0.001) and those who lived with one or two people (*p* = 0.004) had a higher consumption of processed foods than the other groups.

The older adults who were female (*p* = 0.008), those with higher schooling from 9 to 11 years (*p* = 0.030) and ≥12 years (*p* = 0.011), those without partners (*p* = 0.038), and those who lived with one to two people (*p* = 0.017) had a higher consumption of ultra-processed foods than the other groups (Table 1).

Other variables of socioeconomic conditions such as head of family, income in minimum wages, retirement, and work were not significant in the preliminary analyses.

Figure 1 demonstrates the percentage of energy intake of the most consumed foods in the older people according to schooling. Less educated older adults (0 to 4 years and 5 to 9 years) had the highest consumption of rice and beans, whereas the population with the highest education level had the lowest consumption of oils and fats and the highest intake of cheeses, canned goods, sweetened beverages, sweets, and desserts.

Table 2 presents the final model of the association of food consumption according to the degree of processing according to sociodemographic variables. The aspects that were associated with the lowest consumption of in natura or minimally processed foods were the female sex (β = −4.61; *p* = 0.020) and the highest education level of ≥12 years (β = −6.94; *p* = 0.011). Although sex and schooling were associated with lower consumption of in natura or minimally processed foods, schooling demonstrated a higher magnitude with this low intake. As for the culinary ingredients, only the older adults who lived with three or more people had a higher consumption of these foods (β = 2.91; *p* = 0.005).

Regarding the consumption of processed foods, the yellow or indigenous ethnicity (β = 2.91; *p* = 0.030) was negatively associated with the consumption of processed foods. Lower consumption of processed foods was associated with older adults of the yellow or indigenous ethnicity, whereas schooling remained positively associated in the adjusted model (9 to 11 years of schooling, β = 3.22; *p* = 0.027) and for ≥12 years (β = 4.62; *p* = 0.010) presenting its highest magnitude of association. Older people who lived with one to two people had a higher consumption of processed foods (β = 3.85; *p* = 0.007) than those who lived alone.

Regarding the intake of ultra-processed foods, the female sex (β = 3.06; *p* = 0.028) and schooling for ≥12 years (β = 5.50; *p* = 0.001) remained positively associated in the adjusted model, with the highest schooling with the highest magnitude of association. However, older adults who lived with one to two individuals demonstrated a negative association with the consumption of ultra-processed foods (β = −4.08; *p* = 0.020) (Table 2).

## 4. Discussion

In the present study, older women and those with a higher education level had the lowest consumption of in natura or minimally processed foods and the highest consumption of processed and ultra-processed foods. The yellow or indigenous ethnicity presented the lowest magnitude of the consumption of processed foods; older people who lived with three or more individuals had the highest consumption of culinary ingredients, whereas the older people who lived with one to two people had the highest consumption of processed foods and the lowest consumption of ultra-processed foods compared to those who lived alone.

Considering sex, this result corroborates that of another population study that evaluated 1250 adults in Campinas, Brazil, and reported that women had higher consumption of ultra-processed foods with a percentage of energy contribution of 25.2% [23]. In the present study, older women presented an energy contribution from ultra-processed foods of 17.09%.

However, more recent national data from the *Sistema de Vigilância de Fatores de Risco e Proteção para Doenças Crônicas por Inquérito Telefônico* 2020 (VIGITEL 2020—Risk and Protective Factors Surveillance System for Chronic Non-Comunicable Diseases Through Telephone Interview) demonstrated different results, considering that older women consumed more in natura or minimally processed foods and fewer ultra-processed foods than older men. Consumption of five or more servings of in natura or minimally processed foods was higher in older women (38.5%) (>65 years) than in older men (36.5%), and women had lower consumption of five servings or more of ultra-processed foods (7.5% women vs. 10.9% men), respectively [24].

These results of the present study may be explained by several factors that affect the eating habits of older women, especially in the context of food preparation and purchase. With the phenomenon of “feminization of aging” characterized by the increased life expectancy of older women in relation to men, after widowhood, older women may live alone and are more vulnerable to social isolation, especially associated with the departure of children from home and difficulty in accessing or buying food and may directly affect adequate food consumption [25]. In addition, the perception of health by men is lower than that of women since they tend to seek health services less than women [26]. In a study that evaluated the health conditions and use of health services of the Brazilian older population, with 340,000 individuals (≥60 years old) included in the sample of the National Household Sample Survey, 1998, 2003, and 2008, women obtained the predominance in the search for these services throughout these years [27].

A cohort study that used data from the European Prospective Investigation of Cancer—Norfolk study (1996–2002) demonstrated that the lower frequency of social contact with family members was associated with lower variation in the consumption of fruits and vegetables in both sexes [28]. Other studies that have evaluated the consumption of fruits, vegetables, and legumes in older adults report that single individuals who live alone or have social isolation are more susceptible to a decrease in various consumption of foods mainly comprising the in natura or minimally processed group [29].

The results presented here also revealed that the older adults who lived with three people or more people had the highest consumption of culinary ingredients, and those who lived with one to two residents had the highest consumption of processed foods. However, when we evaluated the intake of ultra-processed foods, older people who lived alone consumed more ultra-processed foods than older adults who lived with one to two people. Thus, the hypothesis is confirmed that older people who live with one or more people tend to cook more and use culinary ingredients and processed foods when preparing basic foods for consumption, while the consumption of ultra-processed foods by this population is also lower.

A study by Louzada et al. has reported that older adults who had a partner had the highest chances of having a food consumption that is considered good according to the Healthy Eating Index, consisting of fruits, vegetables and dark green legumes, whole grains, and oils and fats [30]. Another study has revealed that older people with a partner or married had a higher consumption of foods considered healthy than single or divorced men or women who spent more of their income on commercially prepared foods [31]. Thus, interventions to increase or maintain consumption of in natura or minimally processed foods should be encouraged, especially in older women and those who reside alone, in addition to encouraging the maintenance of the activities of purchase and preparation of these foods, with improved social support, and consequently, avoiding the increase in the consumption of ultra-processed foods.

Another interesting result found in the study was the association of yellow or indigenous ethnicity with lower consumption of processed foods compared to older adults who self-declared themselves to be of white ethnicity. This result can be explained by some hypotheses, such as the lack of access to primary healthcare in indigenous communities, and investigating this relationship in depth was not possible. In addition, the number of older adults who self-reported being of these ethnicities remains significantly lower, and more studies should be conducted in order to evaluate the food consumption of this population.

A study that evaluated the consumption of ultra-processed foods between 2008 and 2018 in Brazil reported an average increase of 5.5% over a period of 10 years, being more expressive in black and indigenous people and in population groups with lower levels of education and income. In this same study, a lower consumption of ultra-processed foods by these individuals compared to the white population was observed [32].

Despite these findings on the lower consumption of processed foods of indigenous and yellow ethnicities, other studies have already reported that this difference is demonstrated by the higher consumption of basic foods such as rice and beans by this population. However, the same is not observed with the other items in the group of in natura or minimally processed foods, such as fruits, vegetables, legumes, and seeds.

We also identified an important association between higher education level and a lower intake of in natura or minimally processed foods and a higher intake of processed and ultra-processed foods. This result contradicts some results already described in the literature since more educated individuals possibly value healthy eating more, understand the importance of prevention or control of NCDs, and tend to keep healthier behaviors such as physical activity practice [33]. A literature review concluded that education is the best predictor of sociodemographic conditions and health conditions rather than occupation and income [3]. A study that evaluated data from the NHANES cohort from 2007 to 2012 revealed that the consumption of ultra-processed foods was inversely associated with age and education; those aged > 60 years and with higher education levels had the lowest consumption of ultra-processed foods [17]. Another study also described the same trend in which more educated individuals were less likely to consume ultra-processed foods [34].

However, other studies in Brazil have demonstrated an opposite trend, as in our results. A cohort study ELSA-BRASIL (2008–2010), with 14.378 individuals, also identified that higher income and higher education level were associated with higher consumption of ultra-processed foods [35]. A cross-sectional study that evaluated inequalities in food consumption of the population of 43,554 older people in Brazil, using data from the 2019 National Health Survey (PNS), observed a lower prevalence of bean consumption in more educated older adults, higher consumption of vegetables, legumes, and fruits, and higher intake of soft drinks and sweets [6]. In adults, the same trend is observed. The Epifloripa cohort study revealed that more educated individuals had a higher consumption of not only in natura or minimally processed foods, such as fruits and vegetables, but also ultra-processed foods, including sweets [36].

These changes in food consumption directly impact the lower availability of nutrients and are related to the development of NCDs and the worsening of health status in older adults. The ultra-processed food and beverage industry is responsible for developing aggressive marketing and sales strategies, with prices that are much lower and competitive with foods considered healthier, thereby considerably increasing the availability in supermarkets, access, and consumption by individuals, including those with low incomes [37].

In Brazil, the regulation of ultra-processed foods and beverages is a scenario that is progressing, considering that the industrial sector has a high economic and political power in the country. A study has reported that their direct or indirect influence on the elaboration of public policies resembles that of the tobacco and alcohol industries [38]. Because they have a “corporate political activity”, they could influence public institutions or the decision-making process of public policies. In this way, the industries commonly oppose proposals for regulatory interventions because of their involved interests [37]. However, in a recent publication (July 2023), the Plenary of the National Health Council (CNS) recommended the inclusion of ultra-processed foods and beverages in the category of harmful to health in the tax reform, also reinforcing that the groups that suffer the most from this reality are the black population and the low-income population since 65% of households run by black people live with food restrictions [39].

Another hypothesis that may help explain this result is the increased consumption of meals away from home among more educated older adults, and this factor is associated with increased purchases of ready-to-eat foods and processed and ultra-processed foods and lower consumption of foods considered healthy and rich in nutrients. According to the POF, the prevalence of food intake outside the home in older adults went from 16.1% in 2008–2009 to 19.4% in 2017–2018 [40].

A study that aimed to elucidate the impacts of eating out in relation to obesity and overweight in participants in the 2002–2003 POF has reported that individuals who had meals away from home had a higher prevalence of overweight and obesity, with the highest intake of soft drinks. In older adults, eating out was associated with a higher prevalence of overweight than in younger people [41]. Thus, it is possible to raise the hypothesis that more educated older adults are changing their eating behaviors, opting not to prepare their meals, preferring to buy ready-made foods, or making more meals outside the home, which can impair the intake of important micronutrients to ensure an adequate nutritional status. An increase in overweight and obesity in older adults is a known predictor for the development of NCDs and worsening of health status in this population [42]. Thus, more studies urgently need to affirm the causal relationship between the consumption of processed and ultra-processed foods and the development of chronic diseases in older adults.

This study had some limitations that should be considered when interpreting the results. First, the cross-sectional design made it impossible to evaluate cause and effect, requiring more longitudinal studies in order to measure the evolution of food consumption and its relationship with sociodemographic conditions. Second, this study included a high proportion of women, which may have directly affected the interpretation of the results. However, this bias is expected in this type of study because people who volunteer to participate in health studies are those who have the greatest interest in the subject and, in general, have the highest frequency of availing of health services, especially in the prevention of NCDs. Third, although the method of evaluation of consumption employed (24hFR) is one of the most used methods in population research, data in the present study were analyzed with only one 24hFR. Thus, evaluating the variation of intra-individual consumption and eating habits was not possible. Fourth, social desirability bias may have led participants to underreport consumption of ultra-processed foods such as sweeteners and sweets.

Finally, the NOVA classification of foods was used. Through this classification, measuring the isolated intake of some important nutrients for the discussion, such as the content of sugar, salt, saturated fats, and alcoholic beverages, among others, is not possible. Nevertheless, studies using the NOVA classification could consider changes in the food system, such as the mode of production and distribution of food over the years, and evaluate the nutritional quality of the foods consumed by the population with their health outcomes [7].

Nonetheless, some strengths of our research should also be mentioned. To the best of our knowledge, this study is one of the first to evaluate consumption according to the degree of food processing in older adults. This study comprised a large sample of individuals living in the community and users of primary healthcare. This study also identified the groups that are most vulnerable to the high consumption of ultra-processed foods. Thus, incentive measures could be directed to increase the consumption of in natura or minimally processed foods and foods with better nutritional quality.

## 5. Conclusions

The study identified a lower intake of in natura or minimally processed foods and a higher intake of ultra-processed foods in women and more educated older participants. The yellow or indigenous ethnicity presented the lowest consumption of processed foods; older people who lived with three or more individuals had the highest consumption of culinary ingredients, whereas the older people who lived with one to two people had the highest consumption of processed foods and the lowest consumption of ultra-processed foods. These findings can be used to plan actions and public policies aimed at older individuals in relation to the consumption of foods associated with health, especially in primary and secondary healthcare, to prevent various health outcomes.

## Figures and Tables

**Figure 1 foods-12-04108-f001:**
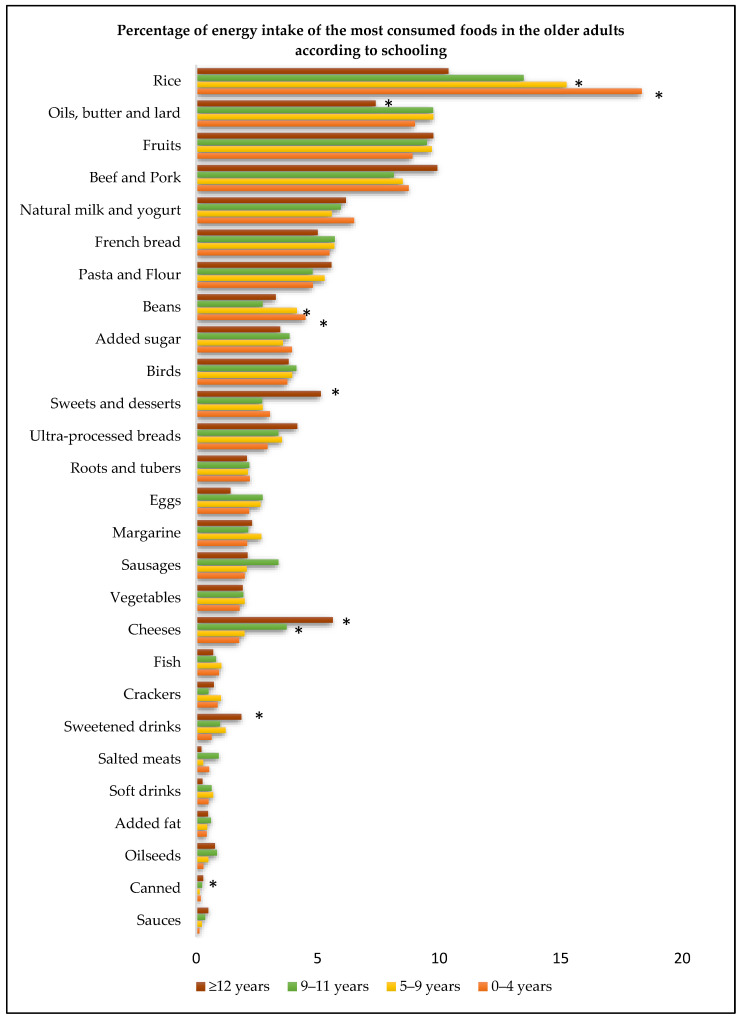
Percentage of energy intake (%) of the most consumed food items according to schooling in community-dwelling older adults. Note: * *p* < 0.05; differences between the means estimated by the Kruskal-Wallis test.

**Table 1 foods-12-04108-t001:** Relative caloric contribution (%) of each food group in relation to the total calorie intake on average (%), standard deviation (SD), and median (%), according to socioeconomic conditions in community-dwelling older adults.

		In Natura or Minimally Processed			Culinary Ingredients			Processed			Ultra-Processed		
Variable	N (%)	Average (%)	SD	Median (%)	Average (%)	SD	Median (%)	Average (%)	SD	Median (%)	Average (%)	SD	Median (%)
Sex													
Male	177 (30)	63.17	15.90	65.30	12.27	6.51	11.70	9.51	9.56	8.22	14.31	13.35	10.65
Female	399 (69)	60.35	15.45	61.56 *	13.17	7.43	11.47	9.02	9.50	7.48	17.09	13.66	14.50 *
Age group													
60–64	159 (28)	60.58	16.90	62.20	12.73	7.05	11.53	8.70	9.45	7.26	17.09	13.96	14.53
65–69	142 (25)	61.07	15.95	63.12	13.26	7.40	11.67	9.00	9.42	7.16	16.09	13.36	13.37
70–74	150 (25)	60.59	15.56	62.50	13.31	7.60	11.61	10.27	10.92	8.82	15.61	13.83	12.38
≥75	125 (22)	62.95	13.58	64.55	12.20	6.50	11.47	8.63	7.71	8.20	16.07	13.29	12.15
Ethnicity													
White	317 (55)	60.5	15.4	62.2	12.9	7.0	11.53	9.9	9.8	8.6	16.2	13.3	16.2
Black/Brown	234 (41)	62.1	15.8	63.9	12.9	7.4	11.6	8.5	9.3	7.3 *	16.0	14.1	16.0
Yellows/Indigenous	22 (4)	62.5	17.4	63.4	12.2	6.6	11.5	6.1	7.8	3.4 *	19.2	14.3	19.2
Years of Schooling													
0–4	295 (51)	63.44	14.76	65.00	12.93	6.98	11.58	8.22	8.89	7.27	15.20	13.94	11.67
5–8	125 (21)	60.81	15.98	61.45 *	13.31	7.25	12.10	8.59	9.33	7.26	16.83	13.40	15.61
9–11	95 (16)	58.00	14.92	60.89 *	13.57	8.03	11.47	10.83	9.50	9.66 *	16.83	11.96	14.60 *
≥12	61 (11)	56.34	18.25	57.13 *	10.85	6.18	10.18 *	12.36	11.76	9.79 *	19.14	14.60	15.75 *
Marital status													
With partner	346 (60)	62.59	14.57	64.00	13.24	7.00	12.64	9.35	9.47	7.88	14.55	12.06	11.90
No partner	230 (40)	59.10	16.95	61.18 *	12.45	7.39	10.77	8.92	9.67	7.55	18.75	15.39	14.94 *
Number of residents													
Alone	94 (16)	60.4	15.2	62.2	12.5	6.9	10.5	7.4	8.2	5.1	18.8	14.2	16.9
1–2	348 (60)	61.7	15.3	63.6	12.7	7.1	11.4	9.8	9.4	8.4 *	15.4	13.2	12.0 *
≥3	134 (23)	60.6	16.8	63.2	13.7	7.5	12.8	8.7	10.4	6.7	16.5	14.1	13.1

Note: * *p* < 0.05; differences between means estimated using the Mann–Whitney U test for two categories and Kruskal–Wallis mean differences test for three or more categories, followed by Dunn’s post-hoc test to evaluate the difference between variables. N, number; SD, standard deviation.

**Table 2 foods-12-04108-t002:** Quantile regression models of the association of sociodemographic variables with food consumption according to the degree of processing in community-dwelling older adults.

Variable	In Natura or Minimally Processed	Culinary Ingredients	Processed	Ultra-Processed
	β Adjusted (CI 95%)	β Adjusted (CI 95%)	β Adjusted (CI 95%)	β Adjusted (CI 95%)
Sex				
Male	–	–	–	–
Female	−4.61 (−8.12; −1.11) *	0.08 (−1.30; 1.46)	0.10 (−2.22; 2.24)	3.07 (0.34; 5.79) *
Age (years)	−0.07 (−0.32; 0.17)	−0.04 (−0.10; 0.09)	0.19 (0.03; 0.35)	−0.03 (−0.22; 0.16)
Ethnicity				
White	–	–	–	–
Black/Brown	1.21 (−2.06; 4.49)	−0.21 (−1.50; 1.08)	−1.88 (−3.90; 0.27)	−1.27 (−3.82; 1.28)
Yellows/Indigenous	0.92 (−7.5; 9.4)	−0.60 (−3.93; 2.72)	−5.95 (−11.33; −0.57) *	0.79 (−5.77; 7.35)
Years of Schooling				
0–4	–	–	–	–
5–8	−2.66 (−8.25; 0.22)	0.44 (−1.18; 2.07)	1.74 (−0.89; 4.37)	2.44 (−0.77; 5.65)
9–11	−3.59 (−8.35; −1.06)	−0.48 (−2.29; 1.31)	3.22 (0.30; 6.13) *	3.15 (−0.40; 6.71)
≥12	−6.93 (−12.89; −1.66) *	−1.48 (−3.64; 0.68)	4.63 (1.13; 8.12) *	5.50 (1.24; 9.77) *
Number of residents				
Alone	–	–	–	–
1–2	0.10 (−4.31; 4.53)	0.99 (−0.75; 2.73)	3.85 (1.03; 6.68) *	−4.08 (−7.52; −0.63) *
≥3	−0.87 (−6.01; 4.3)	2.91 (0.88; 4.93) *	2.12 (−1.15; 5.40)	−1.85 (−5.85; 2.14)

Note: β adjusted for sex, age, schooling, ethnicity, and number of residents’ marital status; * *p* < 0.05; CI, confidence interval.

## Data Availability

The datasets analyzed during the current study are not publicly available but are available from the corresponding author on reasonable request.

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
