# Peer review of "Association of Food Consumption According to the Degree of Processing and Sociodemographic Conditions in Older Adults"

_foods, 2023, doi:10.3390/foods12224108_

Round 1

Reviewer 1 Report

Comments and Suggestions for Authors

The paper addresses a relevant and interesting topic: the elder consumers’ sociodemographic features which affects the healthiness of their diets. However, I see some serious weaknesses both in the research design and presentation. These are summarized below:

2)      The authors do not discuss enough the theoretical background for the choice of the variables introduced in the model. There are some variables that likely influence food preparation and consumption habits that are not included in the model nor discussed qualitatively. Among these, income, the status of worker/retired and the number of hours worked; the number of meals consumed out-of-home, and the number of persons leaving with the interviewee (instead of the marital status that only partially account for this aspect) (see also the discussion in S.4).

3)      The classification of foods according to the degree of processing -adopted in the study- does not seem to be the only relevant aspect that impact the healthiness of the diet. Other features such as: sugar and salt contents; animal proteins and fats, fibers, etc. do play a major role but are not included in the model nor considered adequately in the discussion. Alcohol consumption is neglected as it gives a “negligeable contribution to overall calories intake”; however the diet could be expressed differently than as % calories. In addition, the absolute value of calories eaten should have also been considered.

4)      Information on selection criteria and adhesion are not complete. Is the sample randomly formed?

5)      The description of the sample is incomplete and Incorrect:

a.       the final number of participants in the survey declared (576) is not the result of 611-20-6, as said in lines 83-85.

b.       The distribution of participants according to the focus variables is not fully presented and this affects the possibility to the reader to correctly speculate and understand results.

6)       The discussion of the QR results only revolves around the sign of significant coefficients while the magnitude of the impacts is neglected, while I posit that it is much relevant. Absolute values are always relevant. In addition, what are the base values of consumption in the sample? We need some experts’ evaluation in order to be able to judge whether an x% additional consumption of processed food shall be considered positive/negative for the health…

Comments on the Quality of English Language

1English is bad; there are many grammar and syntax errors, the sequence of tenses is not correct in many points punctuation seems to be often odd.

Author Response

Dear reviewer,

We would like to thank you for the opportunity of submitting the revised version of our manuscript in the Foods. We understand that the manuscript needed some additional changes to be considered for publication.

We would like to thank you for the comments and suggestions. They have been essential to the improvement of our work.

We present the responses to the comments below, and changes are marked in red letters in the text (except minor English editing changes). We also sent to a professional English editing service to improve readability. We hope we have managed to address all issues raised and that the manuscript is now suitable for publication.

Best regards,

The authors

Reviewer 2 Report

Comments and Suggestions for Authors

The paper presents an interesting study on older adults’ self-reported consumption of foods with focus on degree of processing.

Abstract:

L15: remove “that”.

L16-18: Rewrite sentence so it makes grammatical sense. See L63-65.

L22-23: The sentence about statistical analyses is not necessary in the abstract.

Introduction:

L48-50: Please add that the comparison group is less educated.

L52: Here you could include a general paragraph or sentence on why processing could influence health, to provide a smooth transition to the next paragraph.

L61-62: maybe rewrite the sentence from: studies that evaluate food intake in aging are very important to allow educational actions in health for (who? Older adults in general, or older adults with lower education, or?)

Results:

L154: How many (n) were in each group? Please add a column in Table 1 that provide total numbers with percentages in addition to the distribution in percentages for each group.

Table 1: Schooling: 0 a 4 years

Discussion:

L223: remove “in”.

L224: should it be: older women got 17.09% of the energy…. as is presented in Table 1?

L226-232: As you say, the results from VIGITEL contrasts with your results. However, I’m unsure about your discussion of these contrasting findings. On L 232-239, do you mean that the explanation fits your results or those of VIGITEL? Please clarify.

L276-283: Question: What impact do you think price of food has on your findings? It is seen that when the income levels increase in a population, people will change their consumption towards what has traditionally been consumed by the wealthier people. For instance, consumers change from maize to white rice, when they can afford it, despite white rice not being very nutrient rich. Would this be the case of foods with higher degree of processing? That processed foods at some point might be more attractive than foods cooked from scratch?

Please also discuss how or if the high proportion of women influence your results.

References:

All the references are missing DOIs.

Author Response

(The authors gave the same response as above.)
